# Causal Inference for Event Pairs in Multivariate Point Processes

**Tian Gao**
IBM Research
tgao@us.ibm.com

**Dharmashankar Subramanian**
IBM Research
dharmash@us.ibm.com

**Debarun Bhattacharjya**
IBM Research
debarunb@us.ibm.com

**Xiao Shou**
RPI
shoux@rpi.edu

**Nicholas Mattei**
Tulane University
nsmattei@tulane.edu

**Kristin Bennett**
RPI
bennek@rpi.edu

## Abstract

Causal inference and discovery from observational data has been extensively studied across multiple fields. However, most prior work has focused on independent and identically distributed (i.i.d.) data. In this paper, we propose a formalization for causal inference between pairs of event variables in multivariate recurrent event streams by extending Rubin's framework for the average treatment effect (ATE) and propensity scores to multivariate point processes. Analogous to a joint probability distribution representing i.i.d. data, a multivariate point process represents data involving asynchronous and irregularly spaced occurrences of various types of events over a common timeline. We theoretically justify our point process causal framework and show how to obtain unbiased estimates of the proposed measure. We conduct an experimental investigation using synthetic and real-world event datasets, where our proposed causal inference framework is shown to exhibit superior performance against a set of baseline pairwise causal association scores.

## 1 Introduction

It is widely known that the gold standard for effective causal inference is through the use of interventional data such as randomized controlled trials, for measuring the impact of some *treatment* on an *outcome* of interest [30]. However, intervening in the system can often be impractical or even impossible, in which case one must conduct causal analysis using observational data alone.

Observational data in the form of multivariate event streams is common in many domains including health, finance, and retail. Analogous to how an observation in an i.i.d dataset can be viewed as a sample from a joint distribution over a set of random variables, a multivariate event stream can be viewed as a sample from a *multivariate point process* over a set of event labels [10], where the instantaneous rate of any event label depends on the historical occurrences of some subset of the event labels. Modeling, fitting, and predicting future occurrences given a multivariate event stream is an active area in statistics [9, 11, 40], data mining [28, 34], and machine learning [32, 16, 29, 44, 7]. Recent work in this area leverages advances in deep learning [45, 25, 17, 48, 21].

Related work in survival analysis focuses only on pairs of events with continuous covariates [23], involving short event streams with both continuous and discrete variables, and most notably [47, 46] only a single occurrence of the outcome [24]. While a general theory for point process counterfactual inference has been developed [37, 1], practical algorithms focus on continuous covariates and hazard models [39]. Moreover, work in dynamic treatment [26, 18] is for discrete time, motivating work for continuous-time data, which aims to reduce bias or high variance in discretizing time [38, 41, 43]. In

contrast, here we focus on the setting where data is of the form of *long event streams with multiple occurrences* of various types of events, including both the treatment event and outcome event.

Causal inference entails drawing a conclusion about a causal relationship between potential causes and effects; in this work, we focus on causal inference between pairs of event labels observed in event stream data, which differs from standard Granger causal graph learning [12, 49]. Specifically, we pose the following causal inference problem for multivariate point processes: *how can one meaningfully measure the causal relationship between a cause event label $z$ and an effect event label $y$?* Such a causal measure would explain whether the cause label $z$ amplifies, inhibits, or has no impact on the effect label $y$, while taking into account the potential effects of all other event labels $\mathbf{x}$.

This problem deviates from the typical causal inference setting in several ways. First, most methods assume i.i.d. observations; in our setting, events may be correlated across time and therefore it is necessary to make certain independence assumptions in order to identify the effect of interest. Second, most causal inference settings are concerned with the expected value of a single observable outcome such as mortality. Our setting has repeated occurrences of an outcome over time and therefore involves event frequency (or rate), which existing pairwise causal inference models are unable to handle and therefore necessitates new methods. Note that since event label $z$'s historical occurrences could have a complex dynamic effect on another event $y$'s occurrence at any time in a multivariate point process, there are a number of questions to address around the fundamental problem of even defining the notions of treatment, outcome and covariates, before one can study the causal effect of $z$ on $y$.

There are several real-world applications that motivate our work on causal inference with event stream data. For instance, one may be interested in understanding causal relations between events relevant to a diabetic patient, such as meals, insulin intake, exercise and blood glucose level changes [2]. Detecting issues in microservice or other web applications requires detecting which faults or abnormal states cause other downstream faults from system logs [3]. Example causal pairs from other domains are: food shortage and protests, which are recurring event labels in socio-economic datasets [27], and earning call events and stock price jumps, which are recurring event labels in financial datasets [42]. All these examples have an observable outcome that occurs repeatedly, necessitating a new approach.

**Contributions** We make several major contributions: 1) we propose a formalization of causal inference between a pair of event variables in a multivariate point process, 2) we define an average treatment effect (ATE) for multivariate point processes as the effect of having observed treatment in a proximal window, 3) we derive and obtain the equivalence of propensity scores and balancing scores for ATE in multivariate point processes, 4) we propose an inverse propensity score re-weighting procedure for better ATE estimation to adjust for the impact of other covariates, and 5) we conduct a detailed empirical evaluation on both synthetic and real datasets with baselines to show the superior performance of our proposed methods in pairwise causal inference in multivariate point processes.

## 2 Background

A multivariate **event stream** (or **event dataset**) is a sequence of events, $\mathcal{D} = \{\mathcal{D}_i\}_{i=1}^N$, where each event $\mathcal{D}_i$ includes an event label (or synonymously type or variable) and a time-stamp, i.e. $\mathcal{D}_i = (x_i, t_i)$. $x_i$ belongs to the label set $\mathcal{L}$, whose cardinality is $M$, i.e. $|\mathcal{L}| = M$, and $t_i$ is the time of occurrence, $t_i \in \mathbb{R}^+$. We assume a strictly temporally ordered dataset, $t_i < t_j$ for $i < j$, initial time $t_0 = 0$ and end time $t_{N+1} = T$. Let $z, y$ refer to an arbitrary pair of event labels in $\mathcal{L}$.

Multivariate event streams can be regarded as samples from a **multivariate point process (MPP)**, where each label has a counting process [10]. It uses **conditional intensity functions** $\lambda_y(t|\mathcal{H}_t) > 0$ that denote the instantaneous rate of type $y$ given the history up to $t$, i.e. $\mathcal{H}_t = \{(x_i, t_i) : t_i < t\}$. In a multivariate point process, the probability of observing $y$ as the next event at time $t$ is:

$$p_t(y|\mathcal{H}_{t_n}) = \lambda_y(t|\mathcal{H}_t) \exp(-\int_{t_n}^t \sum_{x \in \mathcal{L}} \lambda_x(\tau|\mathcal{H}_\tau) d\tau) \tag{1}$$

where $t_n$ is the most recent event occurrence time before $t$. It can be shown that $\lambda_y(t|\mathcal{H}_t)dt = E[N_y([t, t+dt])|\mathcal{H}_t]$ [10, 33], where $N_y(A)$ is the number of $y$ occurrences in time interval $A$.

Prior work has proposed the notion of **process independence** among event labels' counting processes [11]. The basic idea is that the intensity of one type of event does not depend on certain past events once we know about specific other past events. This is an asymmetric concept, similar to Granger causality. Informally, for sets of labels $\mathbf{x}$, $\mathbf{y}$ and $\mathbf{z}$ s.t. $\mathbf{y} \cap \mathbf{z} = \emptyset$, $\mathbf{x}$ is *process independent* of $\mathbf{y}$ given

**z** when all labels in **x** have conditional intensities that do not functionally depend at any time on the history of labels in **y**, given the history of labels in **z**. Process independence is captured in graphical event models [11, 15], where the conditional intensity for any event label at any time $t$ depends only on historical occurrences of its event label parents in the underlying graph.

A minimal graphical representation can be used to define **direct causes** in multivariate point processes, analogous to those in causal networks. Here we provide a simplified definition of direct causes [11]:

**Definition 1.** *Event label $z$ is a direct cause of label $y$ if $z$ belongs to the minimal set of nodes **u** s.t. $y$ is process independent of all other labels given **u**.*

A minimal set entails that none of its subsets satisfies the property. We highlight that in this work, we are interested in identifying pairwise causal relationships without learning a full graphical model. This is because it can often be computationally intensive as well as unnecessary to learn a full model when one is merely interested in pairwise causality.

## 3 Causal Inference in Point Processes

The Neyman-Rubin potential outcomes causal inference framework estimates the treatment effects between a treated variable and an effect for i.i.d. data [36, 20]. We extend this framework, including *average treatment effects (ATE)*, to study how an event label $z$ affects event label $y$ in a MPP. In this class of models, **treatment** is denoted as $z \in \{0, 1\}$, where 0 is the control and 1 the treatment. The potential **outcome** $y^z$ for each $z$ is the outcome if treatment $z$ is applied. The main difficulty in causal inference stems from the fact that only outcomes from the administered treatments are observed and never any of the other outcomes. Hence it may be viewed as a missing data problem.

We estimate the treatment effects between a treatment variable associated with historical occurrences of $z$ and an outcome variable (or response) associated with $y$'s occurrence, under existing assumptions in the literature. Covariate variables **x** involve historical occurrences of labels other than $z$, i.e. $\mathbf{x} = \mathcal{L} \setminus z$. We assume all variables are observed, hence strong ignorability is satisfied (which we will discuss further in Section 3.2). Next we define treatment, outcome, and covariate variables, and then derive the propensity score for ATE computation.

### 3.1 Defining Treatment, Outcome and Covariates

There are many possible ways to define the treatment, outcome, and covariates in a multivariate point process. We begin with the following general formulation. We use $Z_t$ to represent the treatment variable at time $t$, distinguishing it from the event label $z$. Let $\mathcal{H}_t^z$ be the history of occurrences of event $z$ before $t$, and $\mathcal{K}_t^y$ be the future of event $y$ occurrences at $t$.

**Definition 2.** *General Formulation for Causal Inference: for a pair of event labels $(z, y)$, the **treatment** variable $Z_t$ at time $t$ is a function of $z$'s historical occurrences, i.e. $Z_t = f_z(\mathcal{H}_t^z)$. The **outcome** variable is a function of $y$'s future occurrences at $t$, i.e. $Y_t = f_y(\mathcal{K}_t^y)$. The **covariates** $\mathbf{X}_t$ are a function of historical occurrences of event labels other than $z$, i.e. $\mathbf{X}_t = f_{-z}(\mathcal{H}_t^{-z})$.*

**Treatment and Covariate Models** We need to specify the functions in the general formulation of Definition 2, for instance, we need to summarize historical occurrences $\mathcal{H}_t^z$. Due to *potential multiple (re)-occurrence* of treatment and outcome, the representation of history needs careful modeling based on specific assumptions at each time $t$. We take a simplified yet practical view of treatment, making the **proximal** assumption that *the most recent history is sufficient for causation*. We do this by assuming only a recent time window $[t - w, t)$ for some window $w$ could directly have a causal impact on the outcome at time $t$; such an assumption has been used previously in point process models [7] and provides an efficient and interpretable definition for treatment. It fits many real-world situations where the influence comes primarily from recent history and repeated occurrences have little additional impact on the intensity rate. It also has a strong statistical justification: matching long treatment history could result in infinite sample requirements. We assume covariates are also defined by whether they have appeared at least once in some recent time window.

**Outcome Model** We wish to measure a statistical quantity $f_y(\mathcal{K}_t^y)$ associated with the outcome in a multivariate point process, based on future occurrences of $y$, $\mathcal{K}_t^y$, from time $t$. Many such quantities $f_y(\mathcal{K}_t^y)$ can be used. It is natural to consider either the *instantaneous expected number* of occurrences of the outcome $y$, $E[N_y([t, t + dt])|\mathcal{H}_t]$, in place of $\mathcal{K}_t^y$, or *cumulative expected number*

of occurrences of outcome in a future duration $w_f$, $E[N_y([t, t+w_f])|\mathcal{H}_t]$. Moreover, the expected count $E[N_y([t, t+dt])|\mathcal{H}_t]$ equals $\lambda_y(t)dt$ by definition. Note that at each time, the history can be very different with the proximal assumption, hence the rate at each time in $\mathcal{K}_t^y$ can also be different. The cumulative counts can be similarly written as $\Lambda_y^{w_f}(t) = \int_t^{t+w_f} \lambda_y(t)dt$ for some future duration $w_f$, which can be the total time horizon or infinity, but practically it is more meaningful for a specific duration suitable for the application.

We will focus our analysis with $\lambda_y(t)$ as the outcome in the following, which could be directly extended to outcome $\Lambda_y^{w_f}(t)$. We will compare with both definitions of outcomes in experiments.

Hence, our specific realization of the general formulation results in the following formulation:

**Definition 3.** *Recent History Formulation for Causal Inference: for a pair of event labels $(z, y)$, the binary **treatment** variable $Z_t^w$ at time $t$ is defined by whether or not $z$ has occurred at least once within a window $w$ into the past from $t$. The **outcome** variable is the occurrence rate of the effect label $y$ at time $t$ given the treatment, $Y_t = \lambda_{y|Z_t}(t)$. The **covariates** $\mathbf{X}_t^w$ at time $t$ are a binary vector, depending on whether other event labels have occurred at least once in $[t-w, t), w \in (0, t)$.*

In event streams, it is possible for the cause and effect to be the same label in history, for instance, consider a system with negative self-feedback (eating too much sugar for example). Hence $z$ and $\mathbf{X}_t$ could be identical to or contain $y$, but $z$ and $\mathbf{X}_t$ must be disjoint sets of events.

We summarize the assumptions in this formulation of causal inference: 1) events before $t-w$ have no impact on $y$'s occurrence rate at time $t$. This enables memory in time yet provides a compact representation of history, 2) only the occurrence of $z$ in the window impacts $y$'s rate at time $t$ (regardless of the number of occurrences, hence we do not differentiate between multiple occurrences within $w$), and 3) the specific times of $z$'s occurrences do not further affect $y$'s rate at time $t$ [16, 29, 15]. Such a model can be robust to outliers or noisy historical observations.

## 3.2 Defining Average Treatment Effect

To measure how label $y$ responds to historical occurrences of $z$, average treatment effect (ATE) [36] can be extended to our formulation in MPPs. We define the mean potential outcome $Y_t$ under treatment assignment $Z_t^w = k$ as $\mu_y^k := \frac{1}{T}\int_0^T \lambda_y^k(t)dt$, where $T$ is the maximal time horizon. $\mu_y^k$ has an intuitive interpretation: it is the rate of $y$ given $Z_t^w = k$ in its history $w$, averaged over time horizon $t_0 = 0$ to $T$. Note that $\mu_y^k$ measures the temporal average of instantaneous event occurrence counts under treatment. It is a natural way to aggregate an event's expected occurrence counts in continuous time and captures the event dynamics affected by the proposed treatment.

**Definition 4.** *Average Treatment Effect (ATE) for event pairs is defined as:*

$$ATE = E_{\mathcal{H}_T}[\mu_y^1 - \mu_y^0] = E_{\mathcal{H}_T}[\frac{1}{T}\int_t \lambda_y^1(t) - \lambda_y^0(t)dt] \tag{2}$$

where $\lambda_y^1(t)$ is $\lambda_{y|Z_t}(t)$ at time $t$ if $z$ occurs at least once in $\mathcal{H}_t$ ($Z_t^w = 1$) and $\lambda_y^0(t)$ is $\lambda_{y|Z_t}(t)$ at $t$ if $z$ does not occur in $\mathcal{H}_t$ ($Z_t^w = 0$). The $E_{\mathcal{H}_T}[\cdot]$ is the expectation with respect to the random trajectories over $[0, T]$ as induced by the history dependent MPP. Note that since the rate $\lambda_y^k$ is the instantaneous expected count of occurrences, ATE can also be viewed as a measure of expected count differences, normalized by the length of the time horizon $T$.

For a known multivariate point process, one can find the counterfactual $\hat{\lambda}_{y|Z_t}(t)$ and hence the ATE by looking up the conditional intensity function with the corresponding parental conditions after making an adjustment to the treatment. Specifically, removing all occurrences of $z$ in time period $[t-w, t)$ if the factual is $Z_t^w = 1$ or inserting $z$ at time $t-w$ if the factual is $Z_t^w = 0$.

Theorem 1 provides justification for choosing ATE to measure pairwise causal relations in a MPP:

**Theorem 1.** *If $z$ is not a direct cause of $y$ in a multivariate point process, ATE for pair $(z, y)$ is 0.*

All detailed proofs are in the Appendices. The proof of Theorem 1 directly uses Definition 1 to show that the intensity rate does not change per $z$. The above theorem also implies that if $z$ is not a parent of $y$ in a graphical event model representation of the underlying MPP, the ATE for $(z, y)$ is 0 as in Equation 2 from Definition 4.

To use ATE as defined in Equation 2, there are several assumptions that must hold in order to mimic the randomized trials to truly establish causal relationships. The **ignorability** condition indicates that whether $y$ is 0 or 1 at each time $t$ does not depend on whether $Z_t^w = 1$ or 0 at that time, i.e., $(\lambda_y^1(t), \lambda_y^0(t)) \perp Z_t^w | \mathbf{X}_t, \forall t$ (although the $y$'s intensity rate still depends on $Z_t^w$). Moreover, **overlap** condition states that each time $t$ has a strictly positive chance for $Z_t^w = 1$ to happen given its history, i.e., $0 < P(Z_t^w = 1 | \mathbf{X}_t) < 1$. **Strong ignorability** is often used when both overlap and ignorability hold true. If the assumption does not hold, possibly due to covariate differences in treatment groups, we need to adjust for these effects using propensity scores. This is particularly important in observational event data, where different times $t$ with $Z_t^w = 1$ and $Z_t^w = 0$ may not be directly comparable, because the covariates $\mathbf{X}_t^w$ may not be similar to each other. It can be studies further with the back-door criterion [31].

### 3.3 Defining Propensity Scores

The propensity score is proposed to resolve covariate differences in non-randomized experiments to mimic a randomized study. It is a **balancing score**: conditioned on any balancing score, the distribution of observed covariates will be similar between the treated and control groups. It has been widely used in i.i.d. settings [22, 5, 13, 4].

We define a set of balancing scores, $b(\mathcal{H}_t^{-z})$ ($b_t^*$ for short), where the goal is to make treatment conditionally independent of covariates given the scores: $Z_t \perp \mathcal{H}_t^{-Z} | b(\mathcal{H}_t^{-Z}), \forall t$. Based on the specification provided by Definition 3, we use the recent window assumption and summarize the history $\mathcal{H}_t^{-z}$ with a set of covariates $\mathbf{X}_t^w$. Hence the balancing score would achieve $Z_t^w \perp \mathbf{X}_t^w | b(\mathbf{X}_t^w), \forall t$. The most trivial $b(\mathbf{X}_t^w)$ is $b(\mathbf{X}_t^w) = \mathbf{X}_t^w$. However, to consider the sampling distribution of potential cause $Z_t^w$, many-to-one functions of $\mathbf{X}_t^w$ offer better adjustment, and the coarsest of such a function is the propensity score. We define $X$ is finer than $Y$ (equivalently, $Y$ is coarser than $X$) if $y = f(X)$. "Coarsest" here means that $Y$'s dimension cannot be reduced further.

We use the above defined treatment $Z_t^w$ and covariates $\mathbf{X}_t^w$ and derive the equivalent form of propensity score for MPPs. The goal is to derive the propensity score $e(\mathbf{X}_t^w)$, or $e_t^*$ for short, such that $P(Z_t^w = 1 | \mathbf{X}_t^w, e_t^*) = P(Z_t^w = 1 | e_t^*)$. We know that:

$$P(Z_t^w = 1 | \mathbf{X}_t^w) = \frac{D(Z_t^w = 1; \mathbf{X}_t^w)}{D(\mathbf{X}_t^w)}, \tag{3}$$

where $D(Z_t^w = 1; \mathbf{X}_t^w)$ is the duration that $Z_t^w$ is observed true given observing covariates $\mathbf{X}_t^w$ in the dataset. We define $P(Z_t^w = 1 | e_t)$ similarly as $P(Z_t^w = 1 | e_t^*) = \frac{D(Z_t^w = 1; e_t^*)}{D(e_t^*)}$. $D(Z_t^w = 1; e_t^*)$ can be interpreted as the duration that $Z_t^w$ is observed true, i.e., the total duration for which $z$ occurs in $[t - w, t], \forall t$, in the dataset with the same $e_t^*$ in the relevant preceding windows. To ensure the required equality, we define $e_t^*$ as:

**Definition 5.** *The **propensity score** for all covariates in history $\mathbf{X}_t^w$ at time $t$ is defined as $e_t^* = e(\mathbf{X}_t^w) = P(Z_t^w = 1 | \mathbf{X}_t^w)$.*

Note that $Z_t^w$ at any time $t$ follows a distribution $P(Z_t^w = 1 | \mathbf{X}_t^w)$, which depends on the values of $\mathbf{X}_t^w$. This quantity can be estimated efficiently by counting the duration of $Z_t^w = 1$ with different $\mathbf{X}_t^w$ values, which we will elaborate upon later. The major distinction between the proposed propensity score and that for the traditional i.i.d. case is that $e_t^*$ is time-variant and incorporates all occurrences and non-occurrences of events in a window-viewed history.

We can show that times with the same value $b_t^*$ but different treatment $Z_t^w$ can act as controls for each other, as their expected outcome difference equals the proposed ATE:

**Theorem 2.** *Suppose the treatment event $Z_t^w$ is strongly ignorable and $b_t^*$ is a balancing score. Let $[\mu_y^k | \mathbf{A}] := \frac{1}{T} \int_t \lambda_y^k(t) | \mathbf{A} dt$. Then, $[\mu_y^1 | \mathbf{X}_t^w, Z_t^w = 1] - [\mu_y^0 | \mathbf{X}_t^w, Z_t^w = 0] = [\mu_y^1 | b_t^*] - [\mu_y^0 | b_t^*]$.*

*Proof sketch.* If the treatment is strongly ignorable, then it is also strongly ignorable given any balancing score. Therefore, we can show that the expected differences of both sides are equal. We further can show that treatment $Z_t^w$ and covariates in history $\mathbf{X}_t^w$ are conditionally independent, given the propensity score $e_t^*$, which makes $e_t^*$ a balancing score. Please see the Appendices for exact proofs and further discussions.

It follows that the two-sampling process gives an unbiased estimate of ATE, using pair matching and sub-classification techniques [36] to adjust for the propensity score. However, in practice, pair

matching is difficult to perform given the continuous nature of time $t$ which results in infinite sampling sizes. Subclassification is also difficult when the number of covariate event labels is large, leading to a high sample requirement on $T$.

## 4 Estimating ATE Scores

There are several parameters that need to be provided or estimated in order to adjust for the propensity score and obtain ATE, including the window size for the treatment definition, conditional intensity rate of the treatment, and the outcome conditional or cumulative intensity rates.

**Window Size** $w$ **for the Treatment**: The window size $w$ of the treatment $Z_t^w$ definition is provided as an input to the ATE estimation procedure. Moreover, it is used as the window size for all quantities associated with a window, namely $\mathbf{X}_t^w, Z_t^w, \lambda_{y|Z_t}(t)$, and $P(Z_t^w|\mathbf{X}_t^w)$. If one wants to estimate a window, practitioners should exercise prior knowledge to set $w$ since the treatment effect with different windows will produce different results.

Alternately, one could take a data-driven approach to learning a window for a particular pair of event labels $(z, y)$. One candidate is the heuristic pairwise approach in [7] that chooses from a set of inter-event times in the event stream data. Note that windows that are too short or too long could result in treatments that are always absent or present, respectively, making the ATE computation unsuitable and unstable. There is therefore a trade-off between long (which captures the historical influences as much as possible) and short windows (which captures only the important subset of history). We include more details in the Appendix.

**Propensity Score** $e_t^*$ **Estimation**: Let $D(Z_t^w = 1; \mathbf{x}_t^w)$ be the duration that $Z_t^w$ is observed true in the dataset and that the condition $\mathbf{x}_t^w$ is true in the relevant preceding window $w$. Formally,

$$D(Z_t^w = 1; \mathbf{x}_t^w) = \sum_{i=1}^{N+1} \int_{t_{i-1}}^{t_i} I_{\mathbf{x}_t^w}(Z_t^w = 1, t)dt \tag{4}$$

where $I_{\mathbf{x}_t^w}(Z_t^w = 1, t)$ is an indicator for whether each instantiation of $\mathbf{X}_t^w$, $\mathbf{x}_t^w$, is true and $Z_t^w = 1$ at time $t$ as a function of the relevant windows $w$. Similarly, $D(\mathbf{x}_t^w)$ is the duration that the condition $\mathbf{x}_t^w$ is true in $w$. Hence, $P(Z_t^w = 1|\mathbf{x}_t^w) = \frac{D(Z_t^w=1;\mathbf{x}_t^w)}{D(\mathbf{x}_t^w)}$ and $P(Z_t^w = 1|\mathbf{X}_t^w)$ is a vector of such probabilities with size $|\mathbf{X}_t^w|$.

**Outcome Conditional Intensity Rate** $\lambda_{y|Z_t}(t)$: Given the most recent history view in the treatment definition, we estimate $\lambda_{y|Z_t}(t), \forall 0 \le t \le T$ by using the same recent history formulation. This assumption has been used previously in MPPs, such as in proximal graphical event models (PGEMs) [7] where the rate of occurrence of an event type depends only on whether or not its parents have occurred in the most recent history.

When the parents $\mathbf{U}$ of all nodes $V$ are known, the log likelihood of a MPP represented by a PGEM can be simplified to a function of counts and durations in the data and the conditional intensity rates:

$$\mathrm{logL}(D) = \sum_V \sum_{\mathbf{u}} \left(-\lambda_{v|\mathbf{u}} D(\mathbf{u}) + N(v; \mathbf{u}) \ln(\lambda_{v|\mathbf{u}})\right) \tag{5}$$

where $\mathbf{u}$ is an instantiation of the parent of each variable $v$, $N(v; \mathbf{u})$ is the number of times that $X$ is observed in the dataset and that the condition $\mathbf{u}$ (from $2^{|\mathbf{U}|}$ possible parental combinations) is true in the relevant preceding windows, and $D(\mathbf{u})$ is the duration over the entire time period where the condition $\mathbf{u}$ is true. Formally, $N(v; \mathbf{u}) = \sum_{i=1}^{N} I(l_i = V) I_{\mathbf{u}}^{w_v}(t_i)$ and $D(\mathbf{u}) = \sum_{i=1}^{N+1} \int_{t_{i-1}}^{t_i} I_{\mathbf{u}}^{w_v}(t)dt$, where $I_{\mathbf{u}}^{w_v}(t)$ is an indicator for whether $\mathbf{u}$ is true at time $t$ as a function of the relevant windows $w_v$. We refer the reader to [7] for more details around model learning.

We set $z$ as the parent of $y$ in a PGEM and then can estimate $\lambda_{y|z}^w$ and $\lambda_{y|\bar{z}}^w$, which are intensities when $z$ does and does not occur in the provided proximal window $w$. It is efficient since we only look at two events and only need to estimate intensity rates without learning windows. Once we have these two intensities, we can then look up $\lambda_{y|Z_t}(t)$ by finding the treatment $Z_t^w$ state at any query time $t$.

**Outcome Cumulative Intensity Rate** $\Lambda_{y|Z_t}^{w_f}(t)$: The cumulative intensity is computed as the integral of the conditional intensity over a future window $w_f$, assuming that no other events occur. Again, given a learned point process model, we take a recent history view and compute the integral using

---

**Algorithm 1** Inverse Probability Weighting for Events

---

**Input:** Event data $D$, time horizon $T$
Sample times $t$ from 0 to $T$.
Compute the propensity scores $e_t^*$, $\forall t$, per Equation 3.
Compute $w_t$, $\forall t$, per Equation 6 or stabilized scores.
Train a PGEM $\mathcal{M}_y$ to estimate $\lambda_{y|Z_t}(t)$, $\forall t$.
Predict with $\mathcal{M}_y$ for a given $\mathcal{H}_t$ the outcome $\lambda_y^0(t)$ with $Z_t^w = 1$ and $\lambda_y^1(t)$ with $Z_t^w = 0$.
Compute ATE per Equation 7.

---

estimates for conditional intensities $\lambda_{y|z}^w$ and $\lambda_{y|\bar{z}}^w$. Computing the integral from $t$ to $t + w_f$ is straightforward as these are piece-wise constant conditional intensities. Furthermore, it is only possible for the treatment condition to change from $Z_t^w = 1$ to $Z_t^w = 0$ in the future window, when the condition from the occurrence of event label $z$ changes, but not the other way around.

**Inverse Probability Weighting for Events**: The idea of weighting samples is as follows. If the time units for data involving the treatment $Z_t^w = 1$ group are far fewer than those in the control group $Z_t^w = 0$ in an event dataset, we need to upweight the time units in the treatment group and down-weight those in the control group for better estimation. We use inverse probability of treatment weighting (IPTW) as the weighting scheme, in order to adjust for this imbalance in the population.

We define the weight for the conditional intensity at time $t$ in the treated $Z_t^w = 1$ group as $\alpha_t = \frac{1}{P(Z_t^w=1|\mathbf{X}_t^w)} = \frac{1}{e_t^*}$, and for the control group $Z_t^w = 0$, the weight becomes $\alpha_t = \frac{1}{1-P(Z_t^w=1|\mathcal{H}_t)} = \frac{1}{1-e_t^*}$. Together, the weight for the outcome intensity rate at time $t$ is defined as:

$$\alpha_t = \frac{Z_t^w}{e_t^*} + \frac{1 - Z_t^w}{1 - e_t^*} w \tag{6}$$

Using Equation 6, we propose an inverse probability weighting method for events, detailed in Algorithm 1. We first estimate the propensity score using Equation 3, and then estimate $w_t$ for all $t$ using Equation 6. Then we can choose a PGEM $\mathcal{M}_y$ to predict $\lambda_{y|Z_t}(t)$, for both the factual outcome and counterfactual $\lambda_{y|Z_t}(t)$. Computing their empirical expectation provides an $ATE$ estimate:

$$ATE = E_{\mathcal{H}_T}[\frac{1}{T} \int_{t=0}^{T} \alpha_t \cdot \lambda_y^1(t) - \frac{1}{T} \int_{t=0}^{T} \alpha_t \cdot \lambda_y^0(t)] \tag{7}$$

Equation 7 requires one to compare occurrence rates for all $t$ in $[0, T]$, which includes some epochs with treatment and some without. Since this is a continuous time setting, integration over time is not straightforward. Hence, we propose using a sampling procedure to compute the inner integral over time in Equation 7 as $\frac{1}{S}\sum_{t=1}^{S} \alpha_t \cdot \lambda_y^1(t) - \frac{1}{S}\sum_{t=1}^{S} \alpha_t \cdot \lambda_y^0(t)$, where $S$ is the desired number of epoch samples from $t_0 = 0$ to $T$.

**Corollary 3.** *Inverse probability weighting for events with balancing scores. Under strongly ignorable treatment, the weighted average with inverse probability weighting is unbiased for ATE when weights equal those in Equation 6.*

*Proof Sketch.* The inner integral of $ATE$ can be shown to equal to $[\mu_y^1] - [\mu_y^0]$, and further $[\mu_y^1(t)] = \frac{1}{T}\int_t \frac{Z_t^W \cdot \lambda_y^*(t)}{e_t^*} dt$ and $[\mu_y^0(t)] = \frac{1}{T}\int_t \frac{(1-Z_t^W)\cdot\lambda_y^*(t)}{1-e_t^*} dt$.

We show in Corollary 3 that our ATE definition is an unbiased estimator. Consistency results, however, rely on the estimation results of intensities from specific point process models. For example, when the window sizes are given, PGEM estimator [7] for intensities are consistent.

Since we consider (interval) stationary point process here, we can follow the standard computational procedure to estimate the sample variance of ATE estimator and a sample-based confidence interval for the ATE estimate (for example, results from Theorem 6.2 in [19] can be directly extended to our setting), as the samples are from the same distribution.

**Factual and Counterfactual Treatment Usages** To compute the ATE, we need both a factual and counterfactual treatment for time. For time $t$ with factual $Z_t^w = 1$, the counterfactual $Z_t^w = 0$ treatment is modeled by removing all occurrences of $z$ in $[t - w, t)$. For time $t$ with factual $Z_t^w = 0$, the counterfactual $Z_t^w = 1$ treatment is modeled by introducing one occurrence of $z$ in $[t - w, t)$.

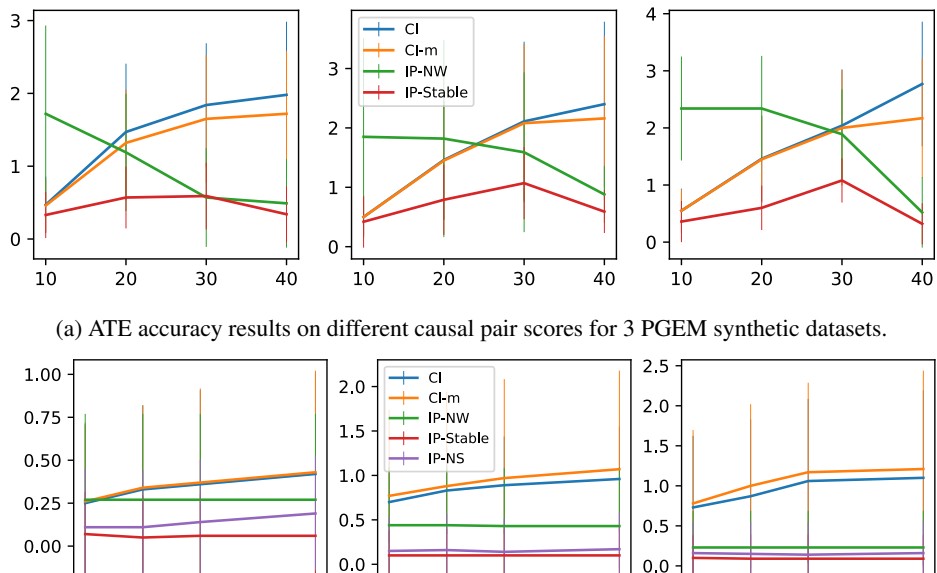

(a) ATE accuracy results on different causal pair scores for 3 PGEM synthetic datasets.

(b) ATE accuracy results on different causal pair scores for 3 Hawkes synthetic datasets.

For ease of computation, we introduce the counterfactual occurrence of $z$ at $t - w$ for instantaneous conditional intensity computation, although its exact location would not impact the outcome due to the proximal assumption. For cumulative intensity outcomes, we introduce the counterfactual occurrences at $t - \delta t$, with $\delta t = \frac{w}{2}$ which is the middle point of the window. Other choices $\delta t$ of introducing the counterfactual treatment are also possible.

**Stabilized IPTW** One common issue with IPTW is that sometimes the propensity scores for some time unit $t$ can get very close to $0$, which indicates $Z$ is extremely unlikely to occur in the window $[t - w, t)$; this is even more likely for continuous time data. Hence, the weights for those $t$ become extremely large, causing unstable estimations. To combat this issue, the stabilized IPTW [35] uses the marginal probability of treatment to counteract such an instability. It is formulated as $\alpha_t = \frac{Z_t^w \cdot P(Z_t^w = 1)}{e_t^*} + \frac{(1 - Z_t^w) \cdot (1 - P(Z_t^w = 1))}{1 - e_t^*}$.

## 5 Empirical Evaluation

Evaluating causal inference algorithms is more difficult than those for prediction tasks since observational datasets rarely contain the ground truth treatment effects. To this end, most experiments in the literature analyze causal models using a synthetic dataset where the ground truth is known [20]. We begin by comparing the ATE estimation performance of our proposed IPTW methods on synthetic event datasets, generated using different parameters. Per standard practice in causal inference literature, we use root mean squared error (RMSE) to measure the ATE accuracy of each method, along with its standard deviation. We first focus on the experiments with the instantaneous conditional intensity outcome. All experiments are done on a machine with 2.9 GHz quad-core CPU.

Since outcome $\lambda_y$ is not directly observable, simple adaptation of ATE from the i.i.d. case would not work. There are not many well-established baselines that use event occurrence rate as the outcome in multivariate point processes for comparison. We adapt two baseline scores from fitting parametric models to the intensity rates from cause-effect associate scores 11[8]: $CI$ (conditional intensity) and $CI_M$, which consider a single parent event and a set, respectively. For the first baseline, we consider the association between a pair of events, $(z, y)$ and assume that the intensity of $y$ only depends on whether or not $z$ has occurred at least once within a specified time window $w$. Hence we define a *conditional intensity* score to estimate the causal effect of $z$ on $y$ as $CI(z, y) = \lambda_{y|z}^w - \lambda_{y|\bar{z}}^w$. We extend this definition to allow for $y$ depending on the historical arrival of a set of events instead of

Table 1: Hits@K results for the diabetes dataset (test set).

| Method | K = 10 | K = 15 | K = 20 |
|--------|--------|--------|--------|
| IP-Stable | 3 | **6** | **7** |
| CI | 1 | 2 | 3 |
| CI$_M$ | **4** | 5 | 6 |

just a single $z$. This means that there are possibly $2^{|\mathbf{X}|+1}$ conditional intensity rates, and we need to aggregate the score for $z$ given all other conditions of the parent set. For our purposes we will use the *mean* of $y$ over all settings. Formally, $CI_M(z, y) = \text{mean}\left(\lambda_{y|z,\mathbf{x}}^w - \lambda_{y|\bar{z},\mathbf{x}}^w\right)$, when $z$ is a parent of $y$ and otherwise the score is 0.

We compare CI scores with three versions of ATE estimation based on our proposed IPTW methods. First, we compute ATE with no weighting as per Equation 2, labeled as IP-NW. Second, we use the proposed IPTW with non-stable weight (IP-NS) with weights as per Equation 6. Last, we use IPTW with stable weight (IP-Stable) to compute ATE.

**Synthetic PGEM Datasets** To test our estimation methods on synthetic data, we first generate event data that adheres to the proximal assumption for the intensity functions [7]. We generate 3 models with different numbers of events, randomly generated graph structures among events, fixed window size of $w = 30$, $T = 2000$, and random intensities between $0.1$ and $0.4$. We use the data and the generated model to obtain the true estimates of $\lambda_{y|Z_t(t)}$ at chosen times $t$ and hence can compute the ground truth ATE. Since we observed that the sample size $S$ of $t$ ($10^3$ to $10^5$) in the ATE estimation does not impact the results much, we use sample size $S = 10^3$ for all our experiments. Shown in Figure 1a on three synthetic PGEM datasets, non-weighted IP-NW performs worse than CI scores when window size $w$ is small but better with large windows. IP-Stable's RMSE is consistently the smallest across datasets and window sizes, sometimes with a 6 times reduction in error from CI scores. We did not include IP-NS in this figure as it has large errors in comparison to other methods. IP-NS with non-stable weight can lead to non-stable estimation. For exact numerical values, please refer to the appendix.

**Synthetic Hawkes Dataset** We also test our approaches on synthetic multivariate Hawkes process datasets using an existing toolbox [6]. We again generate 3 datasets with $30$, $40$, and $50$ event labels, and $w = 15$. We use a fixed base rate $0.016$ and each parental event leads to additive spike of $0.06$ to the base rate, with an exponential decay rate $0.15$. We generate event streams with $T = 2000$. For estimating the counterfactual rate, $\hat{\lambda}_{y|Z_t}(t)$, to compute ATE in the case where there is no treatment event over $[t - w, t)$, we introduce a counterfactual treatment event at time $t - w$. Figure 1b shows the RMSE results of different algorithms. IP-NW outperforms CI scores in all cases. IP-Stable shows the lowest RMSE among all methods, generally achieving 3 to 10 times better accuracy. In addition, the performance of IP-NW and IP-Stable are relatively stable with respect to the window sizes, with minimal changes. For exact numerical values, please refer to the appendix.

**Hybrid Dataset** We also generate a synthetic hybrid dataset that combines proximal graphical event models with additive excitation similar to Hawkes processes with a constant kernel. For further details about the generating process and the results, please refer to Appendices G and H. IP-NW outperforms CI scores in all cases but one, and IP-Stable shows the lowest RMSE in all but two cases.

**Diabetes Dataset** We also test our methods on the diabetes dataset [14] – a real-world dataset which we process into events for meals, exercise activity, insulin dosage and changes in blood glucose measurements for 70 diabetes patients. We treat the assessments in [2] as the ground truth, where an expert provided 11 pairs such that a cause label is more likely to make the effect label occur. Since the assessments are only partial and do not provide the true ATE, we use hits@K among highest absolute estimated ATE values to measure performance in this experiment, which is a popular metric for information retrieval. Specifically, we determine how many of the 11 pairs are recovered by a method's top $K$ absolute scores. The dataset is split into $50\%/50\%$ training/test sets, and optimal window setting is determined on the training set, which is then deployed in the test set for evaluation. $w = \{0.1, 0.3, 0.5, 1\}$ days for all models were considered during training. Table 1 compares the Hits@K across all methods on the test set for $K = \{10, 15, 20\}$, illustrating that IP-Stable is able to recover more of the expert-assessed pairs in two out of three K values.

Table 2: ATE accuracy results with cumulative intensity outcome on diff. causal pairs for synthetic datasets.

| Model | Future Window Size | CCI | $CCI_M$ | CIP-NW | CIP-NS | CIP-Stable |
|---|---|---|---|---|---|---|
| PGEM-1 | 15 | 4.51±0.94 | 4.45±1.52 | 3.20±2.65 | 122.6±60.6 | **3.11±2.42** |
| PGEM-2 | 15 | 3.77±2.96 | 3.98±3.03 | **2.61±0.97** | 538.91±203.36 | 2.87±0.99 |

**Cumulative Intensity Outcome Experiments** We also test our framework with the outcome being the cumulative intensity. The procedure can be done directly by replacing $\lambda_y(t)$ with its cumulative version $\Lambda_y^{w_f}(t)$, computed from a future window $[t, t + w_f]$, as discussed in Section 4. Hence the overall computation becomes $ATE = E_{\mathcal{H}_T}[\frac{1}{T} \int_{t=0}^{T} \alpha_t \cdot \Lambda_y^1(t) - \frac{1}{T} \int_{t=0}^{T} \alpha_t \cdot \Lambda_y^0(t)]$. To adapt the CI baselines to cumulative setting, we compute the cumulative version of CI, $CCI$, with $CCI(t) = CI_t \times w_f$. We also name our methods CIP as the cumulative version of the IP methods. All other computations remain the same. We are given the future window size $w_f = 15$ for all methods. To explore the data-driven approach for learning the window, we do not provide $w$ here and instead let it learn from data [7]. We show the results of ATE for the cumulative intensity outcome with 2 PGEMs in Table 2. The results again confirm better performance of CIP methods over baselines, specially NW and Stable versions. CIP-NS can be unstable as indicated previously.

## 6   Conclusion

We have proposed a framework for pairwise event causality in a multivariate point process and formalized the problem with novel definitions of treatment, outcome, and propensity scores. Our definitions allow for efficient modeling of data where outcomes occur multiple times on the timeline. We estimated the average treatment effect using a propensity score weighting procedure that achieves the best performance against baselines. Our work bridges causal inference with multivariate point processes, showing promising performance in estimating pairwise causal relationships among events. Our framework can be extended in many ways. For example, ATE with multiple causes could be computed if we specify an outcome model given such multiple causes. Our treatment and covariate definitions focus on binary values but could be extended to marked point processes where the event has a real valued measurement, by choosing an appropriate aggregation function and outcome models. Future work could study efficient estimation approaches of ATE without sampling and different historical representations of the treatment. Another direction is a count-based estimand over the current proximal assumption, which would result in multiple treatment value problems.

**Limitations And Societal Impact**: Our method focuses on potentially long event sequences with multiple occurrences of all events. We assume proximal historical influence, along with strong ignorability and no confounding. The usage of causal models should be cautioned, to draw potentially harmful conclusions. In particular, the window sizes in our formulation, which can be subject to human inputs, can induce biases and provide different conclusions with different values.

## Acknowledgement

We thank to anonymous reviewers who have provide helpful comments. This research was partly developed with funding from the Defense Advanced Research Projects Agency (DARPA), under Program No. FA8750-19-C-0206. The views and conclusions contained herein are those of the authors and should not be interpreted as necessarily representing the official policies, either expressed or implied, of DARPA, or the U.S. Government. The U.S. Government is authorized to reproduce and distribute reprints for governmental purposes notwithstanding any copyright annotation therein [1]. XS is supported by supported by IBM AI Horizons Network. Nicholas Mattei was supported by an IBM Academic Award.

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
