# A  Justification on the Proposed Balancing Scores

Following the typical procedure, we show that the typical properties of proposed propensity score hold for our proposed score on event streams.

**Theorem 4.** *Treatment $Z_t^w$ and covariates in history $\mathbf{X}_t^w$ are conditionally independent, given the propensity score $e_t^*$.*

We now show that $e_t^*$ is the coarsest balancing score.

**Theorem 5.** *A function $b_t^*$ is a balancing score, that is $Z_t^w \perp \mathbf{X}_t^w | b_t^*, \forall t$, if and only if $b_t^*$ is finer than $e_t^*$ in the sense that $e_t^* = f(b_t^*)$ for some function $f$.*

Next, we show that adjusting for a balancing score $b_t^*$ is sufficient to produce unbiased estimates of the average treatment effect (Definition 4), if the treatment is strongly ignorable.

**Theorem 6.** *If treatment $Z_t^w$ is strongly ignorable, then it is strongly ignorable given any balancing score $b_t^*$, i.e., if $(\lambda_Y^1(t), \lambda_Y^0(t)) \perp Z_t^w | \mathbf{X}_t^w, \forall t$ and $0 < P(Z_t^w = 1 | \mathbf{X}_t^w) < 1, \Rightarrow (\lambda_Y^1(t), \lambda_Y^0(t)) \perp Z_t^w | b_t^*, \forall t$ and $0 < P(Z_t^w = 1 | b_t^*) < 1$.*

Next, we relate the balancing and propensity score to the estimand of causal event pairs. If a randomly selected time $t_1$ with $Z_{t_1}^w = 1$ is compared to a randomly selected time $t_2$ with $Z_{t_2}^w = 0$, the average difference of the outcome over time is $[\mu_y^1 | Z_t^w = 1] - [\mu_y^0 | Z_t^w = 0]$, where $[\mu_y^k | Z_t^w] := \frac{1}{T} \int_t \lambda_y^k(t) | Z_t^w dt$. However, it is in general not equal to that of Equation 2, as the observed data are not from the marginal distribution of $y$ given covariates with $\lambda_{y|Z_t}(t)$ but rather from the conditional history of $y$ with $\lambda_{y|Z_t}^k(t)$ given $Z_t^w = k$, where $k \in \{0, 1\}$ is a specific instantiation of $Z_t^w$ in a windowed history. Hence, time with the same value $b_t^*$ but different treatment $Z_t^w$ can act as controls for each other, as their expected outcome difference equals the proposed ATE:

**Theorem 7.** *Suppose the treatment event $Z_t^w$ is strongly ignorable and $b_t^*$ is a balancing score. Then, $[\mu_y^1 | \mathbf{X}_t^w, Z_t^w = 1] - [\mu_y^0 | \mathbf{X}_t^w, Z_t^w = 0] = [\mu_y^1 | b_t^*] - [\mu_y^0 | b_t^*]$.*

The proofs of above theorems are listed below.

# B  Proof of Theorem 1

*Proof.* Suppose $z$ is not a direct cause of $y$ in the underlying multivariate point process. The counterfactual case does not change $y$'s conditional intensity rate at any time due to process independence: no matter what changes are made to historical occurrences of $z$, the rate at $t$ does not change, given the historical occurrences of other events. Thus the difference between conditional intensities is 0 at any time and the average over time is also 0. $\qquad\square$

# C  Proof of Theorem 4

*Proof.* We need to show that $P(Z_t^w = 1 | \mathbf{X}_t^w, e_t^*) = P(Z_t^w = 1 | e_t^*))$.

$P(Z_t^w = 1 | \mathbf{X}_t^w, e_t^*)) = P(Z_t^w = 1 | P(Z_t^w = 1 | \mathbf{X}_t^w), \mathbf{X}_t^w) = P(Z_t^w = 1 | \mathbf{X}_t^w)$. Hence, $P(Z_t^w = 1 | X_T^w, e_t^*) = e_t^*, => P(Z_t^w = 1 | e_t^*) = P(Z_t^w = 1 | P(Z_t^w = 1 | X_T^w)) = P(Z_t^w = 1 | X_T^w) = e_t^*$. Hence equality holds as $P(Z_t^w = 1 | \mathbf{X}_t^w, e_t^*) = P(Z_t^w = 1 | e_t^*)$. $\qquad\square$

# D  Proof of Theorem 5

*Proof.* $\rightarrow$: Suppose if $b_t^*$ is finer than $e_t^*$, we show $b_t^*$ is a balancing score. Since $e_t^* = P(Z_t^w = 1 | \mathbf{X}_t^w)$, to show $b(\mathbf{X}_t^w)$ is a balancing score, it is sufficient to show $P(Z_t^w = 1 | b_t^*) = e_t^*$. By definition of $e_t^*$, $P(Z_t^w = 1 | b_t^*) = E[e_t^* | b_t^*]$. Since $b_t^*$ is finer than $e_t^*$ by assumption, $E[e_t^* | b_t^*] = e_t^*$. so that $b_t^*$ is a balancing score.

$\leftarrow$: conversely, suppose $b_t^*$ is a balancing score, but that $b_t^*$ is not finer than $e_t^*$, hence there exist two different time $t_1$ and $t_2$ such that $e_{t_1}^* \neq e_{t_2}^*$ but $b_{t_1}^* = b_{t_2}^*$. However, by the definition of $e_t^*$, $\frac{D(Z_t^w = 1; e_{t_1}^*)}{D(e_{t_1}^*)} \neq \frac{D(Z_t^w = 1; e_{t_2}^*)}{D(e_{t_2}^*)}$, hence $P(Z_t^w = 1 | \mathbf{X}_{t_1}^w) \neq P(Z_t^w = 1 | \mathbf{X}_{t_2}^w)$, which shows $Z_t^w$ and

$\mathbf{X}_t^w$ are not conditionally independent given $b_t^*$ and hence $b_t^*$ is not a balancing score which violates the assumption. Therefore, to be a balancing score $b_t^*$ must be finer than $e_t^*$. $\square$

## E Proof of Theorem 6

*Proof.* If $0 < P(Z_t^w = 1|\mathbf{X}_t^w) < 1$, then it immediately follows $0 < P(Z_t^w = 1|b_t^*) < 1$ by the definition of the balancing score. For the first part, we show that $P(Z_t^w = 1|\lambda_Y^1(t), \lambda_Y^0(t), b_t^*) = P(Z_t^w = 1|b_t^*)$. In Theorem 5, we show that $P(Z_t^w = 1|b_t^*) = e_t^*$, hence we need to show $p(Z_t^w = 1|\lambda_Y^1(t), \lambda_Y^0(t), b_t^*) = e_t^*$.

$p(Z_t^w = 1|\lambda_Y^1(t), \lambda_Y^0(t), b_t^*) = E[P(Z_t^w = 1|\lambda_Y^1(t), \lambda_Y^0(t), \mathbf{X}_t^w)|\lambda_Y^1(t), \lambda_Y^0(t), b_t^*)]$, which equals to $E[P(Z_t^w = 1|\mathbf{X}_t^w)|\lambda_Y^1(t), \lambda_Y^0(t), b_t^*)]$ by assumption, which by definition equals to $E[e_t^*|\lambda_Y^1(t), \lambda_Y^0(t), b_t^*)]$, which equals to $e_t^*$ since $b_t^*$ is finer than $e_t^*$. $\square$

## F Proof of Theorem 7

*Proof.* Suppose a specific covariates value $\mathbf{X}_t^w$ is randomly sampled from the entire time horizon, that is, both the $\mathcal{H}_t^0$ and $\mathcal{H}_t^1$ together, and then two different time $t_1, t_2$ with identical $\mathbf{X}_t^w$ but different treatment $Z_t^w$ values are found[2]. In other words, everything in $\mathbf{X}_t^w$ is identical (events occurring at the same relative window) except $z$. Then in this two-step sampling procedure, the expected difference in outcome is $E_{\mathbf{X}_t^w}[[\mu_y^1|\mathbf{X}_t^w, Z_t^w = 1] - [\mu_y^0|\mathbf{X}_t^w, Z_t^w = 0]]$, where $E_{\mathbf{X}_t^w}$ denotes the expectation with respect to the windowed history covariates in the entire time horizon $T$. If the assignment of $Z_t^w$ is strongly ignorable, then the expected difference in outcome becomes $E_{\mathbf{X}_t^w}[[\mu_y^1|\mathbf{X}_t^w] - [\mu_y^0|\mathbf{X}_t^w]]$, which is equal to the inner integral in ATE definition in Equation 2.

Now suppose we repeat the two-sampling procedure but with a balancing score $b_t^*$ instead of $\mathbf{X}_t^w$ and obtain two treatment observations which have the same value of $b_t^*$ but potentially different values of $\mathbf{X}_t^w$. Given the strongly ignorable treatment assignment, it follows from Theorem 6 that $[\mu_y^1|b_t^*, Z_t^w = 1] - [\mu_y^0|b_t^*, Z_t^w = 0] = [\mu_y^1|b_t^*] - [\mu_y^0|b_t^*]$. Then it follows that

$$E_{b_t^*}[[\mu_y^1|b_t^*, Z_t^w = 1] - [\mu_y^0|b_t^*, Z_t^w = 0]]$$
$$= E_{b_t^*}[[\mu_y^1(t)|b_t^*] - [\mu_y^0|b_t^*]] = [\mu_y^1] - [\mu_y^0]$$

where $E_{b_t^*}$ denotes expectation with respect to the distribution of $b_t^*$ in the entire time horizon.

$\square$

## G Proof of Corollary 3

*Proof.* Since inner integral of $ATE$ can be shown to equal to $[\mu_y^1] - [\mu_y^0]$, we show that $[\mu_y^1(t)] = \frac{1}{T}\int_t \frac{Z_t^W \cdot \lambda_y^*(t)}{e_t^*} dt$ and $[\mu_y^0(t)] = \frac{1}{T}\int_t \frac{(1-Z_t^W) \cdot \lambda_y^*(t)}{1-e_t^*} dt$. First, $\frac{1}{T}\int_t \frac{Z_t^W \cdot \lambda_y^*(t)}{e_t^*} dt = E[\frac{1}{T}\int_t \frac{Z_t^W \cdot \lambda_y^*(t)}{e_t^*} dt|\mathbf{X}_t] = E[\frac{1}{T}\int_t \frac{Z_t^W \cdot \lambda_y^1(t)}{e_t^*} dt|\mathbf{X}_t] = E[\frac{E[Z_t^W|\mathbf{X}_t] \cdot \frac{1}{T}\int_t \lambda_y^1(t)|\mathbf{X}_t dt}{e_t^*}] = E[\frac{1}{T}\int_t \lambda_y^1(t)|\mathbf{X}_t dt] = [\mu_y^1(t)]$. Similarly, the second equality also holds. $\square$

## H Window Learning

one may take a data driven approach to estimate the window between the treatment and outcome, with the objective to maximize the likelihood of the event datasets.

When the parents $\mathbf{U}$ of all nodes $X$ are known, the log likelihood of an event dataset given a PGEM can be written in terms of the summary statistics of counts and durations in the data and the conditional

---

[2]Note that by taking a windowed view of history, we can find such two different times, otherwise no time is going to have identical histories since time 0.

Table 3: ATE accuracy results on different causal pair scores for synthetic datasets

| Model | Window Size | CI | CI$_M$ | IP-NW | IP-NS | IP-Stable |
|---|---|---|---|---|---|---|
| Hybrid-1 | 10 | 0.31±0.17 | 0.31±0.16 | 0.22±0.11 | 0.25±0.17 | **0.20±0.10** |
| | 15 | 0.45±0.26 | 0.48±0.28 | 0.22±0.11 | 0.24±0.14 | **0.20±0.10** |
| | 20 | 0.53±0.30 | 0.56±0.32 | 0.25±0.11 | 0.28±0.20 | **0.20±0.10** |
| | 30 | 0.71±0.45 | 0.72±0.48 | 0.22±0.11 | 0.35±0.20 | **0.20±0.10** |
| Hybrid-2 | 10 | **0.57±0.06** | 0.56±0.07 | 0.59±0.03 | 0.62±0.02 | 0.59±0.03 |
| | 15 | 0.61±0.12 | 0.60±0.15 | **0.59±0.03** | 0.62±0.02 | 0.59±0.03 |
| | 20 | 0.66±0.16 | 0.66±0.22 | 0.59±0.03 | 0.68±0.10 | **0.59±0.03** |
| | 30 | 0.75±0.34 | 0.75±0.36 | 0.59±0.03 | 0.70±0.16 | **0.59±0.03** |
| Hybrid-3 | 10 | 0.65±0.33 | 0.73±0.40 | 0.30±0.12 | 0.29±0.15 | **0.25±0.10** |
| | 15 | 1.02±0.56 | 1.16±0.58 | 0.30±0.12 | 0.32±0.17 | **0.25±0.10** |
| | 20 | 1.28±0.75 | 1.42±0.75 | 0.30±0.12 | 0.40±0.21 | **0.25±0.10** |
| | 30 | 1.54±1.00 | 1.64±0.97 | 0.30±0.12 | 0.45±0.27 | **0.25±0.09** |

intensity rates of the PGEM:

$$\text{logL}(D) = \sum_X \sum_{\mathbf{u}} \left( -\lambda_{x|\mathbf{u}} D(\mathbf{u}) + N(x; \mathbf{u}) \ln(\lambda_{x|\mathbf{u}}) \right), \tag{8}$$

where $N(x; \mathbf{u})$ is the number of times that $X$ is observed in the dataset and that the condition $\mathbf{u}$ (from $2^{|\mathbf{U}|}$ possible parental combinations) is true in the relevant preceding windows, and $D(\mathbf{u})$ is the duration over the entire time period where the condition $\mathbf{u}$ is true. Formally, $N(x; \mathbf{u}) = \sum_{i=1}^{N} I(l_i = X) I_{\mathbf{u}}^{w_x}(t_i)$ and $D(\mathbf{u}) = \sum_{i=1}^{N+1} \int_{t_{i-1}}^{t_i} I_{\mathbf{u}}^{w_x}(t) dt$, where $I_{\mathbf{u}}^{w_x}(t)$ is an indicator for whether $\mathbf{u}$ is true at time $t$ as a function of the relevant windows $w_x$. Note that we have hidden the dependence of the summary statistics on windows $w_x$ for notational simplicity.

From Equation 8, it is easy to see that the maximum likelihood estimates (MLEs) of the conditional intensity rates are $\hat{\lambda}_{x|\mathbf{u}} = \frac{N(x;\mathbf{u})}{D(\mathbf{u})}$. The following theorem uses this to provide a high-level recipe for finding optimal windows for a node given its parents. $N(x)$ denotes counts of event label $X$ in the data.

**Theorem 8.** *[7] The log likelihood maximizing windows for a node $X$ with parents $\mathbf{U}$ are those that maximize the KL divergence between a count-based distribution with probabilities $\frac{N(x;\mathbf{u})}{N(x)}$ and a duration-based distribution with probabilities $\frac{D(\mathbf{u})}{T}$.*

Note that for each time $t \in [0, T]$, there is some one parental state $\mathbf{u}(h_t, w_x)$ that is active. Since the number of such parental states over $[0, T]$ is finite (upper bounded by $\min(2^{|\mathbf{U}|}, 2N)$ and further limited by what the data $D$ and windows $w_x$ allow), this leads to a finite partition of $[0, T]$. Each member in this partition corresponds to some parental state $\mathbf{u}$, and in general, it is a union of a collection of non-intersecting half-open or closed time intervals that are subsets of $[0, T]$. Each member thus has a net total duration, which sums to $T$ across the above partition, and similarly a net total count of the number of arrivals of type $X$. As such, $w_x$ taken with $D$ is equivalent to two finite distributions (histograms) whose support is over the above set of partition members, one each for counts and the durations. The above theorem observes that the optimal $w_x$ is one where the count histogram across the partition members maximally differs from the corresponding duration histogram, as per KL divergence. (All proofs are in the supplementary section.) In informal terms, the windows $w_x$ that lead to MLE estimates for conditional intensities are the ones where the summary statistics of empirical arrival rates differ maximally across the above parental state partition.

The challenge with applying Theorem 8 to the practical issue of finding the optimal windows is that this is in general a difficult combinatorial optimization problem with a non-linear objective function.

Next we provide an upper bound on the optimal window from parent $Z$ to node $X$ regardless of other considerations.

**Theorem 9.** *[7] The log likelihood maximizing window $w_{zx}$ from parent $Z$ to a node $X$ is upper bounded by $\max\{\hat{t}_{zz}\}$, where $\{\hat{t}\}$ denotes inter-event times between two events, which is also taken to include the inter-event time between the last arrival of $Z$ and $T$ (end of the horizon).*

The following theorem shows that when a node has a single parent, one can discover a small number of local maxima from the inter-event times in the data, thereby easily computing the global maximum by exhaustively comparing all local maxima.

**Theorem 10.** *[7] For a node $X$ with a single parent $Z$, the log likelihood maximizing window $w_{zx}$ either belongs to or is a left limit of a window in the candidate set $W^* = \{\hat{t}_{zx}\} \cup \max\{\hat{t}_{zz}\}$, where $\{\hat{t}\}$ denotes inter-event times.*

We use the above theorem in our heuristics for finding the optimal windows and parameters given a parent set.

## I  Synthetic Hybrid Dataset Generation

Given $G(\mathcal{L}, \mathcal{E})$ we associate a homogeneous Poisson process $P_i$ at at some baseline intensity $\lambda_i > 0$ with each node $i \in \mathcal{L}$. A directed edge $(i, j)$ denotes a causal influence exerted by parent $i$ on child $j$ as per the following model. Each arrival of type $i$, say the $k^{th}$ arrival of type $i$ at some time $t_{i,k}$ has the potential of adding a new child Poisson process of type $j$, say $\Delta P_{j,i,k}$, i.e. a child arrival process of type $j$, with intensity $\Delta \lambda_{i,j} > 0$. If realized, this child arrival could start at time $t_{i,k}$ but never lives beyond time $t_{i,k} + w_{i,j}$, i.e. its lifetime is at most $w_{i,j}$. Further, at any given point in time, there is at most one such short-lived child Poisson process of type $j$ due to each edge $(i, j)$, i.e. if there is already a child process, say $\Delta P_{j,i,l}$ that is active at time $t_{i,k}$ for some $l < k$ (e.g. due to a previous $l^{th}$ arrival of type $i$, where $t_{i,l} < t_{i,k}$), then $\Delta P_{j,i,k}$ as described above does not take birth at $t_{i,k}$. Instead, it waits in a first-in-first-out queue until some $\tau < t_{i,k} + w_{i,j}$ when it is both at the head of the queue and the currently active child process of type $j$ has died. It then leaves the queue, takes birth and serves as the currently active child process of type $j$ till time $t_{i,k} + w_{i,j}$, which is when it dies, i.e. its lifetime then is $t_{i,k} + w_{i,j} - \tau$. Note that $w_{i,j}$ may be interpreted as a duration over which parental causal influence of type $i$ over type $j$ is active. And the above model manifests such causal influence in terms of an elevated arrival rate for the child $j$ whenever there is an active influence due to any parental historical occurrence of type $i$. The net intensity for each node $i$ at any given time $t$ is then given as:

$$\lambda_i^{\text{net}}(t) = \lambda_i + \sum_{j \in \text{Pa}(i)} \Delta \lambda_{i,j} \delta_j(t) \tag{9}$$

where $\delta_j(t)$ is a binary indicator of active parental influence from type $j$, i.e., $\delta_j(t) = 1$, if $\exists k$ such that $t < t_{i,k} + w_{i,j}$.

## J  Exact Numerical Results for Synthetic Datasets

We include the exact numerical results in Table below.

## K  Additional Results on Hybrid Dataset

We also generate a synthetic point process dataset using a hybrid approach that combines the idea of proximal graphical event models and additive excitation similar to Hawkes processes with a constant kernel. Specifically, each parent that is active over a proximal window (as in PGEMs) contributes a corresponding constant additive excitation to the base rate. For further details about the generating process, please refer to the appendix. We generate 3 models with 30, 40, and 50 event labels with ground truth window $W = 15$. We generate data using the same $T = 2000$. The bottom section of Table 3 shows the RMSE results of different algorithms. IP-NW outperforms CI scores in all cases but one, and IP-Stable shows the lowest RMSE in all but two cases.

Table 4: ATE accuracy results on different causal pair scores for synthetic datasets

| Model | Window Size | CI | CI$_M$ | IP-NW | IP-NS | IP-Stable |
|---|---|---|---|---|---|---|
| PGEM-1 | 10 | 0.47±0.15 | 0.46±0.13 | 1.72±1.47 | 2.66±2.09 | **0.33±0.10** |
|  | 20 | 1.47±0.88 | 1.32±0.54 | 1.19±0.65 | 22.8±6.82 | **0.57±0.18** |
|  | 30 | 1.84±0.72 | 1.65±0.77 | **0.57±0.46** | 54.5±12.2 | 0.59±0.21 |
|  | 40 | 1.98±1.01 | 1.72±0.74 | 0.49±0.37 | 4.93±3.20 | **0.34±0.15** |
| PGEM-2 | 10 | 0.50±0.12 | 0.50±0.11 | 1.85±2.76 | 1.54±1.79 | **0.42±0.19** |
|  | 20 | 1.46±1.00 | 1.45±1.00 | 1.82±2.75 | 19.1±33.9 | **0.79±0.35** |
|  | 30 | 2.11±1.80 | 2.08±1.78 | 1.59±1.81 | 24.5±11.3 | **1.07±0.37** |
|  | 40 | 2.40±1.93 | 2.16±1.91 | 0.88±0.23 | 26.4±2.51 | **0.59±0.14** |
| PGEM-3 | 10 | 0.55±0.15 | 0.55±0.14 | 2.34±0.83 | 1.20±0.59 | **0.36±0.13** |
|  | 20 | 1.46±0.57 | 1.45±0.57 | 2.34±0.85 | 12.8±5.09 | **0.60±0.15** |
|  | 30 | 2.04±0.97 | 2.00±1.02 | 1.89±0.62 | 95.8±10.5 | **1.08±0.15** |
|  | 40 | 2.77±1.19 | 2.17±1.10 | 0.52±0.38 | 5.71±2.36 | **0.32±0.13** |
| Hawkes-1 | 10 | 0.25±0.21 | 0.26±0.21 | 0.27±0.25 | 0.11±0.12 | **0.07±0.06** |
|  | 15 | 0.33±0.24 | 0.34±0.23 | 0.27±0.25 | 0.11±0.11 | **0.05±0.06** |
|  | 20 | 0.36±0.30 | 0.37±0.30 | 0.27±0.25 | 0.14±0.13 | **0.06±0.06** |
|  | 30 | 0.42±0.35 | 0.43±0.35 | 0.27±0.25 | 0.19±0.11 | **0.06±0.06** |
| Hawkes-2 | 10 | 0.70±0.78 | 0.77±0.94 | 0.44±0.43 | 0.15±0.15 | **0.10±0.08** |
|  | 15 | 0.83±0.92 | 0.88±1.13 | 0.44±0.44 | 0.16±0.17 | **0.10±0.09** |
|  | 20 | 0.89±1.00 | 0.97±1.24 | 0.43±0.43 | 0.14±0.14 | **0.10±0.09** |
|  | 30 | 0.96±1.11 | 1.07±1.23 | 0.43±0.44 | 0.17±0.17 | **0.10±0.10** |
| Hawkes-3 | 10 | 0.74±0.80 | 0.78±0.84 | 0.23±0.21 | 0.16±0.15 | **0.10±0.08** |
|  | 15 | 0.87±0.92 | 1.00±1.04 | 0.23±0.21 | 0.15±0.16 | **0.09±0.09** |
|  | 20 | 1.06±1.05 | 1.17±1.25 | 0.23±0.21 | 0.14±0.16 | **0.09±0.09** |
|  | 30 | 1.10±1.19 | 1.21±1.51 | 0.23±0.21 | 0.16±0.16 | **0.09±0.09** |