# OpenReview forum: "Causal Inference for Event Pairs in Multivariate Point Processes"
_NeurIPS.cc/2021/Conference — NeurIPS 2021 Poster_

### Official Review · Reviewer_oC9j · 2021-07-16

**Rating:** 5
**Confidence:** 3

**Summary:**

This paper proposes a causal inference framework for a multivariate point process model. The key idea is to form a series of binary processes Z_t and X_t using a moving window [t-w,t] to indicate whether processes Z and X have at least one event in the window [t-w,t].  An average treatment event (ATE) is defined and propensity scores are proposed for unbiased estimation of ATE. The authors use synthetic experiments and real-world datasets to demonstrate how the proposed method work compared to baseline approaches.

**Limitations And Societal Impact:**

1. Line 154, I have some doubts on the assumption (3) "the specific times of z’s occurrences do not further affect y’s rate at time t.". This is at least not the case for Hawke's process. I wonder how many existing point process models satisfy this assumption. Can you give some examples?

2. Line 240, if the dimension of X_t^w is high, wouldn't the propensity estimates be very unstable and very high dimensional?

3. In Corollary 3. only an unbiased estimator for the ATE is given. But to do causal inference, one would need some distributional results for ATE. Can this be done? Otherwise, I do not see much usage of the proposed method in real applications.

**Main Review:**

The paper is fairly well written and the concepts proposed are clear and new. I would say that the work is quite original in terms of conceptual development.

**Time Spent Reviewing:**

24

---

> ### Author Response · Authors · 2021-08-10
> **Thank you for your review**
>
> We thank you for your time reviewing our work and providing valuable comments.
>
> 1. **Examples that satisfy the mentioned assumption**:  Many event models such as proximal graphical event model (PGEM) [Bhattacharjya et al, 2018],  piece-wise constant intensity model (PCIM)  [Gunawardana et al., 2011], CPCIM [Parikh et al., 2012], and more generally Timescale graphical event model (TGEM) [Gunawardana et al., 2016] all follow this assumption.
> Gunawardana et al. [2016] show that this family of graphical event model is a universal approximator for multivariate temporal point processes.
>
> 2. **High dimension**:  High covariate dimension estimation could indeed be an issue but please note that this is a pervasive issue. As typical in standard graphical models, if the parental set is large, parameters may require a lot of samples for accurate estimation without further modeling assumptions.
>
> 3. **Distributional results for ATE**: Since we consider (interval) stationary point process here, we can follow the standard computational procedure to estimate the sample variance of ATE estimator and a sample-based confidence interval for the ATE estimate (for example, results from Theorem 6.2 in [Imbens \& Rubin, 2015] can be directly extended to our setting), as the samples are from the same distribution. We will add such a discussion in the revision.
>
> > Imbens, G., & Rubin, D. (2015). Causal Inference for Statistics, Social, and Biomedical Sciences: An Introduction. *Cambridge: Cambridge University Press*.
>
> > A. Gunawardana, C. Meek, and P. Xu (2011). A model for temporal dependencies in event streams. In *
> Proceedings of Advances in Neural Information Processing Systems (NIPS)*, pages 1962–1970, 2011.
>
> > A. P. Parikh, A. Gunawardana, and C. Meek (2012). Conjoint modeling of temporal dependencies in event
> streams. *Proceedings of Uncertainty in Artificial Intelligence Workshop on Bayesian Modeling Applications*.
>
> > A. Gunawardana and C. Meek (2016). Universal models of multivariate temporal point processes. In
> *Proceedings of the Nineteenth International Conference on Artificial Intelligence and Statistics
> (AISTATS)*, pages 556–563, 2016.
>
> > Bhattacharjya, D., Subramanian, D., & Gao, T. (2018). Proximal graphical event models. Advances in Neural Information Processing Systems, 31, 8136-8145.

---

### Official Review · Reviewer_ktAc · 2021-07-16

**Rating:** 7
**Confidence:** 3

**Summary:**

The paper provides tools to estimate average treatment effect (ATE) between event pairs. The authors first define the quantity for multivariate point processes together with the necessary concepts from Neyman–Rubin causality, then they derive a propensity score in this setting. Finally, they give a method for estimating the ATE and evaluate the method on synthetic and real world datasets.


**Limitations And Societal Impact:**

The work precisely define causal assumptions defining the limits of applicability.

**Main Review:**

The work precisely define causal assumptions of the method: ignorability and overlap. The method do not require causal sufficiency, unobserved confounders are allowed as long as the covariates form a proper adjustment set for their effect. However it will ignore indirect effects mediated by these covariate variables.

What does it mean that $\mathcal{K}$ is $y$'s future occurrences at $t$? Later we see that $Y_i$ is actually equal the instantaneous expected number of occurrences  OR the cumulative expected number of occurrences of the outcome. At this point the entity $\mathcal{K}$ is somewhat mysterious.

Line 131 (typo): y is the effect/outcome not a treatment

Line 180: Ignorability is misinterpreted here. Ignorability means that the counterfactual outcomes are independent from the actual outcome given the covariates. It is actually telling that the current value of Z in the real world have no extra information to determine the counterfactual outcomes given the knowledge of the covariates. It is closely related to the back-door criterion, see Pearl: Causality, 2009, Chapter 3; 11.

Line 198: What do you mean by "the coarsest of such a function is the propensity score" ?

Line 204: missing : given observing covariates $\mathbf{x}_t^w$




**Time Spent Reviewing:**

4

---

> ### Author Response · Authors · 2021-08-10
> **Thank you for your review**
>
> We thank you for your time reviewing our work and providing valuable comments.
>
> 1. **$\mathcal{K}$ and $f(\mathcal{K})$**: We use $\mathcal{K}^y_t$ to represent all future occurrences of event $Y$ at $t$, and $f_y(\mathcal{K}^y_t)$ is some function over the future occurrences. We introduce the notation to be general in our definition, and both instantaneous and cumulative expected number of occurrences are specific instantiations of $f_y(\mathcal{K}^y_t)$. We will clarify this in the revision.
>
> 2.  **Ignorability and "Coarsest"**: Thanks for pointing this out.  We directly follow Rosenbaum and Rubin [`83] and use their definitions. We will clarify that "coarsest" means that $y$'s dimension cannot be reduced further. We will also clarify the statement on ignorability and draw the connections to the back-door criterion per suggestion.
>
> 3. We thank the reviewer for pointing out the typos and will fix them as suggested.

---

> > ### Comment · Reviewer_ktAc · 2021-08-25
> > **Correction**
> >
> > Thank you for your answers.
> > A correction from my side:
> > "Ignorability means that the counterfactual outcomes are independent from the actual ~~outcome~~  **causative factor X** given the covariates." Your definition is perfectly correct. The connection with back-door criterion is valid as stated.

---

### Official Review · Reviewer_CcPK · 2021-07-16

**Rating:** 6
**Confidence:** 4

**Summary:**

This paper proposes a model for estimating the average causal effect in point processes where the estimand of interest is the effect of an event occurring within a window on the rate parameter of a point process. The authors use the formulation commonly used for inverse propensity scores of generalized treatments (e.g., from Robins, Imai). The authors describe the properties of the propensity score as a balancing weight by leveraging known results and advocate for the use of stabilization of inverse propensity scores to reduce variance as in Robins(2000). The weights are then used within an outcome model which is a point process model. Empirical results show promising performance compared to a range of alternatives.


**Limitations And Societal Impact:**

Any method for causal inference can be mis-applied an have negative societal impacts, however I don't think there is anything specific to this work that is unaddressed and should be acknowledged.

**Main Review:**

I think this is an interesting and important problem area, and the authors provide what I feel to be a reasonable approach. While the paper is on the lighter side in terms of theoretical contributions, I think the approach is novel and provides a nice addition to the literature.

My largest concerns are as follows:
- The authors could do more to tie the proposed approach to the larger literature. While the point process application is novel (to my eyes), there is a large literature on propensity scores and balancing weights for general treatments in the statistics literature which should be referenced in order to provide proper context.

- It's unclear to me how much is being sacrificed by considering the quantity "event happens any number of times within the window" over "number of times an event is observed within the window". It would seem to me that these are different estimands, however it is not clear when we would prefer this "event happens at all" estimand. Can the authors provide intuition on this point?

- It would be nice to have consistency / convergence results for the proposed estimator.

- It's unclear how one would get confidence intervals from the proposed approach for the estimates.

- The authors offer a heuristic for choosing the window size for the estimate. Intuitively, it would seem that smaller window sizes should result in greater precision. Can the authors provide intuition for the observed behavior which appears either flat or bell shaped?

- It would be much more convincing if the authors ran experiments on any of the motivating domain examples. As-is it is difficult to reason over the true efficacy of the proposed approach on the synthetic domains.

Some smaller points:
- The authors consistently conflate propensity scores with balancing weights. Propensity scores are balancing weights under correct specification but the converse is not necessarily true. Given the growing literature on balancing weights this distinction is important to adhered to for the benefit of the reader.

- The citation to Zubizaretta (2015) on line 298 is incorrect. Zubizarreta’s paper concerns the construction of balancing weights, not stabilized propensity scores. I believe the citation for stabilized IPTW should be Robins (2000).

**Time Spent Reviewing:**

4

---

> ### Author Response · Authors · 2021-08-10
> **Thank you for your review**
>
> We thank you for your time reviewing our work and providing valuable comments.
>
> 1. **Related literature and wording on propensity score**: Thanks for pointing them out. We will cite more related papers from standard and more recent literature. We will clarify the exact wording as suggested and cite related work.
>
> 2. **Proximal assumption**: The assumption around an estimand for "event happens at least once in some recent window" fits many real-world situations where the influence comes primarily from recent history and repeated occurrences have little additional impact on the intensity rate.
> It has been argued for in prior work and seems to fit many real-world event streams well [Bhattacharjya
> et al., 2018].
> The reviewer likely suggested a count-based estimand over the current proximal assumption, which would result in multiple treatment values. We think this could be an interesting future direction to study.
>
> 3. **Consistency results**: We show in Corollary 3 that our ATE definition is an unbiased estimator. Consistency results, however, rely on the estimation results of intensities from point process models. For example, when the window sizes are given, PGEM estimator $\mathcal{M}_y$ for intensities  are consistent. We will add such a discussion.
>
> 4. **Confidence interval**:  Sample-base confidence intervals for the proposed ATE could be computed similar to typical procedures in i.i.d. data  [Imbens \& Rubin, 2015], since we obtain empirical samples  to compute outcomes and ATE at different times.
>
> 5. **Window learning**:
> The heuristic for window learning is a data-driven approach that helps determines windows that are potentially domain-appropriate. It is not necessarily true that smaller window sizes are always better, since it may ignore relevant historical influence and effectively make the point process homogeneous. Also, note that windows that are too short or too long could result in treatments that are always absent or present, respectively, making the ATE computation unsuitable and unstable.  There is therefore a trade-off between long (which captures the historical influences as much as possible) and short windows (which captures only the important subset of history).
>
>
> 6. **Example domains**: We have provided results on a real diabetes dataset in the paper. Real datasets rarely have the ground truth measure for ATE, hence it is difficult to judge the accuracy of various methods. This is why we emphasized synthetic datasets in our experiments.
>
> > Bhattacharjya, D., Subramanian, D., & Gao, T. (2018). Proximal graphical event models. Advances in Neural Information Processing Systems, 31, 8136-8145.
>
> > Imbens, G., & Rubin, D. (2015). Causal Inference for Statistics, Social, and Biomedical Sciences: An Introduction. *Cambridge: Cambridge University Press*.

---

### Official Review · Reviewer_BtNu · 2021-07-19

**Rating:** 8
**Confidence:** 3

**Summary:**

This paper describes a method for the causal analysis of long streams of
event data that can be modeled as multivariate point processes (MPPs).
The paper defines the average treatment effect (ATE) between pairs of
events in MPPs that occur within a window, and derives how to estimate
the ATE as the expected difference of average outcome rates.  When
covariates differ or assumptions are violated, this estimate may be
biased, but adjusting for the propensity score yields an unbiased
estimator, which the paper does through inverse probability of treatment
weighting (IPTW) (including a stabilized version).

Experiments on data from known, synthetic MPPs (proximal graphical event
models and Hawkes processes) show that the proposed IPTW estimation
method more accurately recovers the ATE than baseline conditional
intensity scores.  Similarly, experiments on a diabetes dataset show
that stable IPTW recovers more true causal pairs at 2 out of 3 levels of
recall compared to the conditional intensity scores.  These results
extend to using the cumulative rather than the conditional intensity
rate.


**Ethical Concerns:**

No ethical concerns.


**Limitations And Societal Impact:**

The authors didn't really address the limitations or potential negative
impacts of their work.  (However, many limitations are implicitly
addressed in that one can deduct them from the content, but it's always
nice to have important things explicitly addressed.)  The only thing I
can quickly think of other than general, routine cautions about causal
models and confounding (which I don't miss here), is some discussion of
the window size and its consequences.  (But some experiments showed no
effect.)  I agree with the authors (in the checklist) that there is
unlikely to be societal impact.


**Main Review:**

Overall
-------

Overall, I think this is an excellent paper.  In particular, it balances
well the presentation of high level concepts and motivations while
providing rigorous, supporting details.  The framework of causal
inference in multivariate point processes (MPPs) it develops could play
a central role in establishing a comprehensive methodology for causal
inference with event sequence data.


Originality
-----------

While all of the pieces of this paper have been well established, this
paper combines them in a new way by deriving the standard causal
inference measure (ATE) -- and how to estimate it well -- for a class of
models (MPPs) that are very relevant to practical applications.  The
paper situates its contributions clearly in comparison to survival
analysis, hazard models, and Granger causality.  From the meticulous
nature of the paper, I expect related work is adequately cited, but I
don't know the literature in this area thoroughly enough to assuredly
make that judgment.


Quality
-------

The submission is very meticulously written and is technically sound as
far as I have investigated.  (I did not check the supplement nor the
proofs therein.)  Statements are well-cited, and experimental evidence
supports the claims of effectiveness.  I don't have any complaints about
the methodology or the completeness.  The authors do not include an
explicit evaluation of strengths and weaknesses, but they have defined
their scope well enough to allow those to be inferred.

My main suggestion is to use figures to illustrate the results rather
than just listing them in tables.  Yes, I know it takes a lot more work,
but the end result is so much easier for our feeble human brains to
interpret.  (If you think the specific numbers are important, then put
them in the supplement, perhaps as a data file.)  This alone brings my
evaluation down one point.

I also suggest including proof sketches in the main body of the paper as
a general rule of thumb (if it is accepted you get an extra page), but
the theoretical results are as expected, so I'm less concerned compared
to unexpected / interesting results.


Clarity
-------

This paper is one of the clearest I have ever reviewed.  Bravo!  (I have
no suggestions.)


Significance
------------

I think other researchers are quite likely to build on this work in
helping to establish a framework of causality for MPPs.  Since much of
the observational data we collect is event-based, I believe causality
with MPPs is quite important, and this paper could be a foundational
contribution to the area.


Details
-------

These are just some things I noticed; they are not comprehensive.

78-79 It would be nice to have a citation here, if only for the
new-comers.  (Where is it shown that ...?)

218 Is the square brackets and pipes notation, '[...|...]', meant to be
expectation?  That's what the text implies, but there's no 'E'.  Please
fix or clarify.

411 "hawkes" -> "Hawkes"

421 "2010" -> "2020"


**Time Spent Reviewing:**

4.8 hours

---

> ### Author Response · Authors · 2021-08-10
> **We thank you for your time reviewing our work and providing valuable comments.**
>
> We appreciate the reviewer for the positive comments.
>
> 1. **Figure**: Thanks for the suggestion. We will include a figure instead of Table 1 to show the results and move the table to the appendix.
>
> 2. **Proof sketch**: Thanks for the suggestion. We will include some proof sketch and/or high level discussion for each theoretical result.
>
> 3. We will fix other details (including adding $E$), and add the citation as suggested.
>
> 4. We will expand our discussion around the limitation of our formulation,  which will include explicit assumptions made.

---

### Public Comment · ~Aniq_Ur_Rahman1 · 2025-01-28
**Code Availability**

In the Checklist (3a) the authors promised to release the code of Github. Could the authors please point to the public Github repository link.

Thank you.

---

### Decision · Program_Chairs · 2021-09-27

**Decision:**

Accept (Poster)

**Comment:**

Causal inference in point processes is a novel and open area, and the reviewers agreed that this is a useful contribution both methodologically and from an applied perspective. There are a number of changes that you should definitely make to improve the manuscript.